# Development and Characterization of a Recombinant galT-galU Protein for Broad-Spectrum Immunoprotection Against Porcine Contagious Pleuropneumonia

**DOI:** 10.3390/ijms26083634

**Published:** 2025-04-11

**Authors:** Jia-Yong Chen, Yi Deng, Jiale Liu, Xin Wen, Yu-Qin Cao, Yu Mu, Mengke Sun, Chang Miao, Zhiling Peng, Kun Lu, Yu-Luo Wang, Xizhu Chen, Siyu Pang, Dan Wang, Jiayu Zhou, Miaohan Li, Yiping Wen, Rui Wu, Shan Zhao, Yi-Fei Lang, Qi-Gui Yan, Xiaobo Huang, Senyan Du, Yiping Wang, Xinfeng Han, San-Jie Cao, Qin Zhao

**Affiliations:** 1Research Center for Swine Diseases, College of Veterinary Medicine, Sichuan Agricultural University, Chengdu 611130, China; chenjiayong@stu.sicau.edu.cn (J.-Y.C.); dengyi@stu.sicau.edu.cn (Y.D.); liujiale@stu.sicau.edu.cn (J.L.); 2024203038@stu.sicau.edu.cn (X.W.); yuqincao0627@163.com (Y.-Q.C.); pymuyu@163.com (Y.M.); sunmengke1026@163.com (M.S.); 202001774@stu.sicau.edu.cn (Z.P.); lukun@stu.sicau.edu.cn (K.L.); 2024303131@stu.sicau.edu.cn (Y.-L.W.); 2024303120@stu.sicau.edu.cn (X.C.); 202000826@stu.sicau.edu.cn (S.P.); 2024203037@stu.sicau.edu.cn (D.W.); 202200986@stu.sicau.edu.cn (J.Z.); limiaohan@stu.sicau.edu.cn (M.L.); wyp@sicau.edu.cn (Y.W.); wurui1977@sicau.edu.cn (R.W.); zhaoshan419@163.com (S.Z.); y_langviro@163.com (Y.-F.L.); yanqigui@126.com (Q.-G.Y.); rsghb110@126.com (X.H.); senyandu@sicau.edu.cn (S.D.); yipingwang@sicau.edu.cn (Y.W.); 2Sichuan Science-Observation Experimental Station of Veterinary Drugs and Veterinary Biotechnology, Ministry of Agriculture and Rural Affairs, Chengdu 611130, China; 3Key Laboratory of Animal Disease and Human Health of Sichuan Province, Science & Technology Department of Sichuan Province, Chengdu 611130, China; hanxinf@163.com; 4International Joint Research Center of Animal Disease Control and Prevention, Science & Technology Department of Sichuan Province, Chengdu 611130, China

**Keywords:** porcine contagious pleuropneumonia (PCP), *Actinobacillus pleuropneumoniae* (APP), rgalT-galU, antigen, immunoprotection effect

## Abstract

Porcine contagious pleuropneumonia (PCP), caused by *Actinobacillus pleuropneumoniae* (APP), is a highly contagious disease that leads to significant economic losses in the swine industry. Current vaccines are ineffective due to the presence of multiple serotypes and the absence of a predominant seasonal serotype, underscoring the need for vaccines with broad-spectrum protection. Previous studies identified galT and galU as promising antigen candidates. In this study, we expressed and characterized a soluble recombinant galT-galU protein (rgalT-galU) from the pET-28a-galT-galU plasmid. The protein, with a molecular weight of 73 kDa, exhibited pronounced immunogenicity in murine models, as indicated by a significant elevation in IgG titers determined through an indirect ELISA. This immune response was further corroborated by substantial antigen-specific splenic lymphocyte proliferation, with a stimulation index of 51.5%. Immunization also resulted in elevated serum cytokines levels of IL-4, IL-12, and IFN-γ, as detected by cytokine assays. Vaccination with rgalT-galU provided immunoprotection against three predominant APP strains (APP1, APP5b, and APP7), achieving protection rates of 71.4%, 71.4%, and 85.7%, respectively. It also effectively mitigated pulmonary lesions and neutrophil infiltration, as verified by histopathological and immunohistochemical analyses. These results indicate that rgalT-galU is a promising candidate for developing cross-protective subunit vaccines against APP infection.

## 1. Introduction

Porcine contagious pleuropneumonia (PCP) is a highly infectious disease caused by the bacterial pathogen *Actinobacillus pleuropneumoniae* (APP), primarily affecting the porcine respiratory tract [1]. PCP has remained one of pigs’ most serious respiratory diseases for decades since its first discovery in 1957 [2]. APP can trigger acute porcine pleuropneumonia, a highly contagious disease that can suddenly affect vulnerable pigs, leading to severe symptoms and high mortality rates [3,4]. While pigs suffering from chronic infections caused by APP can be treated with antibiotics and other pharmaceutical interventions, the outcomes of such treatments are frequently suboptimal. This deficiency often results in decreased growth rates and reduced feed efficiency among affected pigs, thereby significantly negatively impacting the economic viability of pig farming and leading to considerable financial losses for the global swine industry [5]. Consequently, PCP is a significant bacterial disease that poses a threat to the sustainable development of the global swine industry [6].

APP has been classified into 19 serotypes, and the dominant serotypes differ across countries and regions, potentially causing piglets to be infected with multiple serotypes [7,8]. China is the world’s largest pig-raising country; the dominant APP strains currently circulating in China are APP1, APP5b, and APP7 [9]. APP encompasses numerous serotypes, yet the cross-protection among these serotypes remains inadequate. A critical concern is the potential for APP to co-infect hosts alongside other pathogens, often necessitating antibiotic intervention to treat infected swine [10]. The rise in antibiotic resistance poses a substantial challenge to the effective management of infectious diseases [11]. The complexity of antibiotic therapy is exacerbated by the urgent clinical demands and the delays associated with obtaining antimicrobial susceptibility results from diagnostic laboratories [12]. Currently, vaccination is still frequently utilized as a strategy to prevent the disease. It is important to note that the prevalence of APP serotypes varies across regions, influencing vaccination strategies. For example, serotypes 2, 5, and 6 are prevalent in Denmark, while serotypes 7, 4, and 2 are common in Spain. Therefore, regional serotype distribution plays a crucial role in determining the most effective vaccination approach [13]. Given these variations, it is essential to tailor vaccination programs to the specific serotype prevalence and infection dynamics of each region to achieve optimal disease control. Commercially available vaccines, including inactivated bacterins and subunit vaccines, face significant clinical limitations, such as undesirable side effects and serotype-specific protection [14,15,16]. For instance, Aptovac targets serovars 2 and 6, while Serkel PleuroAP covers serovars 1 through 5. In Europe, vaccines like Neumosun primarily target serovars 2, 4, and 5 but show limited effectiveness against Asian-prevalent strains. While these vaccines provide partial protection against their specified serotypes, they demonstrate inadequate cross-protection against heterologous strains [17,18]. Consequently, the development of a novel genetically engineered subunit vaccine is of paramount importance, given the substantial economic burden that APP places on the swine industry. The advantage of genetically engineered vaccines is that they can be designed to target specific antigens, thereby improving the specificity and effectiveness of immune responses [19]. Recent studies have highlighted the potential of using particular proteins from APP as components of a subunit vaccine. For instance, numerous APP virulence factors and proteins exhibit robust immunogenicity and offer substantial immune protection. Multi-component vaccines, composed of multiple protein recombinants, represent an effective strategy for enhancing immune efficacy. The fusion expression of nFliC with APP-RTX–toxin (Apx) II enhances both humoral and cellular immunity, resulting in vaccine efficacy reaching 60% to 80% [9]. Bac-sub, comprising three proteins (ApxIA, ApxIIA, and ApxIIIA), exhibits minimal side effects compared to conventional vaccines while boosting antibody levels against virulent strains of APP serotypes 1, 5, and 7 [16]. The Apa2H1 protein, a trimeric autotransporter adhesin, has been shown to activate dendritic cells and induce protective immune responses in mice. Immunization with Apa2H1 produced antigen-specific antibodies and improved survival rates in infected mice, demonstrating its potential as a vaccine candidate [20]. A combinatorial vaccine that includes inactivated bacterin and recombinant protoxins has also been developed, showing high protective efficacy against APP infection in mice and pigs. This vaccine induced strong immune responses and provided significant protection against various serovars of the pathogen [16].

Most known subunit vaccines offer adequate protection against specific serotypes. However, enhancing cross-protection to ensure comprehensive coverage across serotypes is critical in PCP prevention.

In pigs that have survived either natural or experimental infections with APP, the host’s internal environment—characterized by factors such as limitations in NAD or iron—can induce the bacterial expression of certain conserved antigens, including Apx toxin and TbpB protein. These antigens, which may be conserved across various serotypes, potentially contribute to cross-serotype immunoprotection [21,22]. However, achieving robust cross-serotype protection likely necessitates synergistic interactions among multiple antigens, highlighting the need for further validation of their expression mechanisms. Existing studies predominantly focus on specific APP serotypes (e.g., serotype 10) and controlled growth conditions, yet they lack data on cross-serotype protection, which is crucial for practical field applications. Additional research gaps have been identified as follows: First, there is a deficiency in long-term immunity assessment. The duration of protection, the necessity for booster vaccinations, and the rate of immune decline have yet to be evaluated. Second, there is a lack of comprehensive safety evaluations. Current data are insufficient to adequately assess adverse effects or impacts on pig health and growth. Third, there is a need for the validation of field applicability. The efficacy of vaccines under real-world conditions, including co-infections, stress, and farm management practices, has not been confirmed. Future research should prioritize these areas to advance the development of broadly protective vaccines against APP. Previously, through the application of in vivo-induced antigen technology (IVIAT) and bioinformatics tools, we successfully identified galactose-1-phosphate uridyltransferase (galT) and UTP-glucose-1-phosphate uridylyltransferase (galU) as potential candidate antigens for vaccine development [23]. Among various candidates, galT and galU were identified as promising antigens due to their conserved nature across different *Actinobacillus pleuropneumoniae* (APP) serotypes. Notably, galT conferred strong protection against APP5b, while galU offered only partial protection but exhibited a superior immunostimulatory capacity, promoting splenic lymphocyte proliferation and upregulating IFN-γ, IL-2, and IL-4 levels. Despite these findings, the cross-serotype protective efficacy of these antigens remains unverified, and the potential synergy of a rgalT-galU fusion protein has not yet been explored [24]. *galT* and *galU*, members of the *gal* gene cluster, play critical roles in LPS core biosynthesis, which are closely linked to APP virulence [25,26]. galT governs membrane transport systems and boosts extracellular polymeric substance biosynthesis, bolstering bacterial freeze-drying resilience [27]. Furthermore, galU contributes to bacterial adhesion and motility. Mutation in galU significantly reduces LPS mutant adherence relative to the parental strain, indicating its relevance in APP adhesion [28].

In this study, the rgalT-galU protein was prepared and assessed in a murine model to evaluate its conservation, immunogenicity, and protective efficacy against virulent strains of three major serotypes: APP1, APP5b, and APP7. These findings suggest promising vaccine candidates for preventing and controlling APP and provide theoretical references for developing cross-protective vaccines against various APP serotypes.

## 2. Results

### 2.1. Bioinformatics Analysis of the rgalT-galU

The complete coding sequence (CDS) of rgalT-galU is 2055 base pairs, encoding 685 amino acid (aa) residues, resulting in a relative molecular mass of 77 kDa. The theoretical isoelectric point (pI) is calculated as 6.19, suggesting a net negative charge for rgalT-galU under standard physiological conditions. The instability index, computed at 38.35, implies stability for the recombinant protein. The bioinformatic analyses of rgalT-galU provided valuable insights into its physicochemical and structural properties. The Kyte–Doolittle hydropathy plot revealed that rgalT-galU is predominantly hydrophilic, suggesting good solubility (Figure 1A). The antigenic index predicted by DNAStar Protean was higher for rgalT-galU compared to galT and galU individually, indicating that the fusion protein could have enhanced immunogenic potential (Figure 1B). The structural modeling results revealed a well-folded protein, characterized by a balanced distribution of α-helices and β-strands, which supports its stability and potential to expose epitopes (Figure 1C). The SignalP 6.0 prediction confirmed the absence of a signal peptide, suggesting that the protein remains cytoplasmic (Figure 1D). Secondary structure analysis indicated that rgalT-galU contains 21.61% α-helix, 15.62% β-strand, and 62.77% other structural elements, implying that it has a flexible conformation suitable for immune recognition (Figure 1E). These data provide a reliable basis for expressing rgalT-galU.

In the conservation analysis, the nucleotide and amino acid sequences of the *galT* and *galU* genes from serotypes 1–18 of APP exhibit significant conservation. The nucleotide sequences of the *galT* gene show more than 93% similarity across serotypes, while their amino acid sequences display more than 97% similarity (Figure 1F, Appendix A). Similarly, the *galU* gene sequences are highly conserved, with nucleotide similarities exceeding 95% and amino acid similarities above 98% (Figure 1F, Appendix A). The significant conservation suggests that these genes play critical roles in the bacterium’s physiology and potentially its pathogenicity, as the rgalT-galU potentially confer cross-protection against different serotypes of APP.

The comprehensive bioinformatics analysis conducted herein suggests that the rgalT-galU fusion protein exhibits significant potential as a candidate molecule for cross-serotype vaccine development, attributable to its favorable physicochemical properties, structural advantages, and gene conservation.

### 2.2. Identification of rgalT-galU

The *galU* gene was successfully amplified (Figure 2A) and ligated with pET28a-galT through homologous recombination, resulting in the pET28a-galT-galU recombinant plasmid, confirmed by MEGA sequence alignment. We introduced the recombinant plasmid into *E. coli* BL21 (DE3) that is to be expressed and verified by SDS-PAGE and Western blotting. Western blotting was then performed using mouse anti-His-tag monoclonal antibody and anti-galT-galU polyclonal antibody. The SDS-PAGE results revealed that rgalT-galU was detectable in the soluble fraction after cell lysis (Figure 2B). The Western blotting analysis confirmed that the expressed protein (73 kDa) exhibited the expected molecular weight (Figure 2C,D).

### 2.3. Immunogenicity of rgalT-galU

The 96-well ELISA plates were coated with rgalT-galU, rgalT, and rgalU. Subsequently, specific IgG antibodies elicited by these proteins in immunized mice were quantified using ELISA. Notably, no antibodies were detected in the serum from the PBS control group, whereas the immunized groups exhibited a significant increase in IgG levels. The rgalT-galU effectively stimulates humoral immunity, enhancing antibody production (Figure 3A).

We coated 96-well ELISA plates with three serotypes of APP antigens to determine antibody titers via ELISA. No antibodies were detected in the serum from the PBS control group. Specifically, antibody titers in the rgalT-galU-immunized group were 1:600 for APP1, 1:600 for APP5b, and 1:800 for APP7, indicating a robust and consistent humoral immune response across the different serotypes. In contrast, the commercial vaccine group exhibited antibody titers of 1:400 for APP1, 1:400 for APP5b, and 1:800 for APP7 (Figure 3B). Notably, the antibody titers in the rgalT-galU group exceeded those observed in the commercial vaccine group, where responses showed variability among the different serotypes. These results suggest that the rgalT-galU fusion protein may elicit a more uniform and cross-protective immune response across multiple APP serotypes, while the commercial vaccine exhibited a comparatively stronger immune response to some serotypes and weaker responses to others. Serum samples from each mouse group were assessed for IFN-γ, IL-4, and IL-12 levels using sandwich ELISA (Figure 3C). The rgalT-galU group exhibited significantly elevated cytokine levels compared to the PBS group. Specifically, IFN-γ levels were halved, IL-4 levels remained stable, and IL-12 was upregulated twofold relative to the commercial vaccine group. These results indicate that rgalT-galU effectively enhances both humoral and cellular immune responses, warranting the further assessment of additional related cytokines to determine its advantages over the commercial vaccine.

To assess the cellular immune response elicited by rgalT-galU immunization in mice, we investigated the proliferation of splenic lymphocytes. In the PBS control group, splenic lymphocytes proliferated only upon ConA protein stimulation. Conversely, splenic lymphocytes from the rgalT-galU-immunized mice exhibited proliferation when stimulated with rgalT-galU, rgalT, rgalU, and ConA. Following ConA protein stimulation, the proliferation rate of splenic lymphocytes in the rgalT-galU group reached 53.1%. This rate was 51.5% after rgalT-galU stimulation, exceeding 40.7% and 38.2% observed with rgalT and rgalU stimulations, respectively (Figure 3D). These results show that the immunogenicity of rgalT-galU is markedly enhanced compared to that of rgalT and rgalU. We will next focus on comparing rgalT-galU with the commercial vaccine.

### 2.4. Protective Efficacy Provided by rgalT-galU

Twelve hours post-challenge, some mice exhibited clinical symptoms, including lethargy, huddling, rapid breathing, closed eyes, and disheveled fur. All mice in the PBS group died within 24 h post-challenge. In the rgalT-galU-immunized group, the protection rates against APP1, APP5b, and APP7 were 71.4%, 71.4%, and 85.7%, respectively, compared to 85.7%, 71.4%, and 71.4% in the commercial vaccine group (Figure 4A). These results indicate that rgalT-galU provides immune protection against APP1, APP5b, and APP7, comparable to the commercial vaccine.

After the APP1 challenge, mice in the PBS group displayed extensive alveolar wall thickening, inflammatory cell infiltration, and hemorrhage. The commercial vaccine and rgalT-galU markedly attenuated lung tissue damage, resulting in localized alveolar rupture and minimal inflammatory cell infiltration (Figure 4B). Following the APP5b challenge, the PBS group showed widespread alveolar thickening, inflammatory cell infiltration, hemorrhage, necrotic epithelial cells, and eosinophilic material. The commercial vaccine group presented milder alterations but with significant vascular congestion, while the rgalT-galU group showed minimal lesions (Figure 4C). Following the APP7 challenge, the PBS group exhibited irregular bronchial epithelial cells and alveolar hemorrhage. In contrast, the commercial vaccine group had an irregular cell arrangement with minimal eosinophilic material, and the rgalT-galU group exhibited minor inflammatory cell infiltration and slight bleeding (Figure 4D). Overall, rgalT-galU immunization effectively mitigated lung damage across all three APP serotypes, potentially surpassing the protective efficacy of the commercial vaccine.

After challenging mice with APP1, APP5b, and APP7, we assessed lung neutrophil infiltration using immunohistochemistry (Figure 5A). In the PBS group, the IOD values were 5.4 × 10^4^, 5.2 × 10^4^, and 6.8 × 10^4^, indicating pronounced neutrophil infiltration. The commercial vaccine group displayed IOD values of 9 × 10^3^, 2.5 × 10^4^, and 1.2 × 10^4^, significantly reducing pulmonary inflammation. The rgalT-galU-immunized mice had IOD values of 2.8 × 10^4^, 2.4 × 10^4^, and 2.6 × 10^4^ (Figure 5B). Neutrophil infiltration in the rgalT-galU-immunized lung tissue was reduced compared to the PBS group. Although the reduction was less marked than in the commercial vaccine group, rgalT-galU provided notable protection, effectively mitigating pulmonary inflammation.

## 3. Discussion

PCP, caused by APP, is a highly significant swine production disease responsible for substantial economic losses worldwide. The current vaccines, primarily whole-cell and subunit vaccines, have shown varying degrees of efficacy against different serovars of the pathogen. Still, they often fail to provide broad-spectrum protection across all serovars [29,30].

Our prior research identified six in vivo-induced antigens with confirmed immunoprotective properties [24]. Among these, *galT* and *galU* were found to be upregulated during infection and conserved across different serotypes. The *gal* gene cluster plays a pivotal role in the biosynthesis of the lipopolysaccharide (LPS) core, and both *galU* and *galT* are critical components of galactose metabolism, significantly contributing to the pathogenic mechanisms of various bacteria [23,31]. Bacterial pathogenicity mainly depends on the ability to sense and adapt to host environmental signals, enabling pathogens to adjust gene expression to better acclimate upon invasion. Antigens induced in vivo and critical to virulence mechanisms are often highly conserved across strains, making them promising candidates for vaccine development [32,33].

Building on prior research, we undertook a comprehensive genomic analysis of 18 representative APP serotypes and conducted antigenicity predictions utilizing DNASTAR Lasergene (Version 17; DNASTAR Inc., Madison, WI, USA; https://www.dnastar.com). Our analyses revealed that the *galT* and *galU* genes uniquely encompass all serotypes 1–18. Our findings indicate that the nucleotide and amino acid sequences of *galT* and *galU* are highly conserved across APP serotypes 1–18. Specifically, *galT* gene exhibits nucleotide similarity exceeding 93% and amino acid similarity surpassing 97%, while the *galU* gene shows even greater conservation, with nucleotide similarity exceeding 95% and amino acid similarity surpassing 98%. These results corroborate recent studies highlighting the significance of conserved antigenic targets in developing broad-spectrum vaccines. For instance, Liu et al. demonstrated that fusion proteins incorporating conserved bacterial epitopes enhance cross-reactivity across serotypes, a principle integral to our design strategy [34]. Additionally, studies have shown that recombinant proteins generated by fusing different sequences can enhance immunoprotection [15].

The molecular weight of 73 kDa for rgalT-galU suggests successful dimerization or post-translational modifications, consistent with observations by İncir et al. and Girish et al. on recombinant bacterial protein expression in *E. coli* systems [35,36]. The robust IgG response observed in murine models corroborates the immunogenic potential of rgalT-galU. Precisely, the marked elevation in antibody titers and specificity aligns with the findings of Mohammadreza et al., who reported that multi-epitope fusion antigens induce stronger humoral immunity than single-antigen formulations [37]. Furthermore, the splenic lymphocyte proliferation rate (51.5%) and cytokine profiles (IL-4, IL-12, IFN-γ) indicate a balanced Th1/Th2 immune response, a critical feature for effective antibacterial immunity, as highlighted by Tania [38]. Research indicates that elevated levels of IFN-γ enhance macrophage activation, thereby augmenting their pathogen-killing efficacy [39]. IFN-γ, IL-4, and IL-12 are pivotal cytokines within the immune system, each contributing significantly to the modulation of humoral and cellular immune responses. IFN-γ is predominantly secreted by activated T lymphocytes and natural killer cells, facilitating the activation of macrophages and enhancing the function of antigen-presenting cells, thereby augmenting the host’s defense against bacterial pathogens [40]. IL-4 is primarily produced by T helper 2 (Th2) cells and is instrumental in the activation of B lymphocytes and the subsequent production of antibodies, particularly immunoglobulin E (IgE), which is crucial in mediating responses to parasitic infections and allergic reactions. IL-12, secreted by antigen-presenting cells such as macrophages and dendritic cells, promotes the differentiation of T helper 1 (Th1) cells and the production of IFN-γ, thereby playing a vital role in the cellular immune response [41]. In the context of bacterial infections, the interaction among these cytokines is of significant importance. Empirical evidence indicates that interleukin-12 (IL-12) augments the body’s cellular immune response by promoting the differentiation of T-helper 1 (Th1) cells and stimulating the production of interferon-gamma (IFN-γ), thereby effectively counteracting bacterial infections [42]. Concurrently, IFN-γ further enhances the body’s anti-infective capabilities by activating macrophages and augmenting their bactericidal functions. Conversely, the role of interleukin-4 (IL-4) is more intricate, as it may inhibit Th1 cell activation under certain conditions, potentially exerting a negative impact on the immune response to bacterial infections [43]. Our research findings demonstrated that, in comparison to the PBS group, the rgalT-galU group exhibited a significant increase in the levels of IFN-γ, IL-4, and IL-12. The elevated concentrations of these cytokines suggest that rgalT-galU enhances both humoral and cellular immune responses, which are essential for adequate protection against APP.

The observed immunoprotection rates of 71.4%, 71.4%, and 85.7% against APP1, APP5b, and APP7 underscore the potential of rgalT-galU as a cross-protective antigen. Consistent with previous research, the increased protection rate against APP7 indicates a substantial serotype-independent immunity [29]. The *galT* and *galU* proteins are integral to LPS biosynthesis, which is crucial for bacterial virulence and immune evasion [25,26]. Previous studies have demonstrated that recombinant fusion proteins, which combine conserved bacterial epitopes, can enhance cross-reactivity and offer protection against multiple serotypes. For example, a survey of APP found that a recombinant tandem epitope vaccine, which targeted conserved antigens, provided cross-protection against APP challenge in mice [15]. Similarly, multi-epitope vaccines targeting highly conserved bacterial antigens have shown broad protection against diverse serotypes of *Streptococcus suis* [34]. Our research findings demonstrate that rgalT-galU elicits robust immune responses in murine models and confers protective effects against three distinct serotypes: APP1, APP5b, and APP7. This finding suggests that rgalT-galU confers protective effects that transcend specific serotypes, underscoring its potential to provide broader immunological protection. Future investigations will aim to evaluate additional APP serotypes to ascertain whether rgalT-galU can offer broader immunological protection across diverse serotypes.

The ability of rgalT-galU to mitigate pulmonary lesions and neutrophil infiltration further supports its efficacy as a vaccine candidate. This is consistent with the findings from other studies that have demonstrated the role of subunit vaccines in reducing tissue damage caused by bacterial infections [44,45]. Our study’s significant reduction in lesion severity suggests that rgalT-galU provides immunological protection and limits APP infection’s pathological consequences.

Current commercial vaccines for PCP include inactivated and subunit vaccines. However, these vaccines are limited by side effects and insufficient cross-protection. Most existing subunit vaccines provide adequate protection only against specific serotypes. For instance, Quan K et al. used the *E. coli* outer membrane vesicles (OMVs) to deliver the galT protein, which enhanced immunogenicity and improved protection against APP5 [46]. Similarly, Kim et al. demonstrated that the *E. coli* heat-labile enterotoxin B subunit, fused with a neutralizing epitope of ApxIIA exotoxin, formed functional pentamers that enhanced systemic and mucosal immune responses, offering better protection against APP2 [47]. A key challenge in PCP prevention is improving cross-protection to achieve broader serotype coverage. This study utilized a commercial vaccine and PBS as positive and negative controls, respectively. Throughout the immunization process, antibody titers in the rgalT-galU group steadily increased, following a typical antibody response pattern. The rgalT-galU induced a stronger humoral immune response, resulting in higher antibody production and more effectively activated cellular immune responses, significantly upregulating cytokine levels in the serum. This study demonstrates that rgalT-galU effectively addresses the limitations of inactivated and single-subunit vaccines, significantly reducing the clinical symptoms of APP infection in mice and improving survival rates. This approach provides a theoretical foundation for developing a comprehensive and safe vaccine for PCP.

While our results are encouraging, further studies are needed to evaluate the efficacy of rgalT-galU in swine models, which is the natural host of APP [16]. Additionally, the long-term durability of the immune response and the potential for adjuvant optimization should be explored [48,49]. Although adjuvants were not tested in this study, we used MONTANIDETM Gel 02 ST, a novel aqueous polymer-based adjuvant known for its high safety profile and minimal side effects. This adjuvant is commonly used in vaccine formulations for animals, such as pigs and cattle, to significantly enhance immune responses. Previous studies have demonstrated that this adjuvant, when combined with DNA vaccines against BoHV-1, can improve immune protection, and its use in conjunction with inactivated PRRSV vaccines has enhanced cross-protection across genotypes [50,51]. While our study did not focus on adjuvants, future research in swine models will include the screening and evaluation of adjuvants to optimize vaccine efficacy. Integrating rgalT-galU into existing vaccine formulations or its use as a standalone vaccine could significantly improve the control of PCP in the swine industry.

The development of the recombinant fusion protein rgalT-galU represents a significant advancement in addressing the longstanding challenge of achieving cross-serotype immunoprotection against APP.

## 4. Materials and Methods

### 4.1. Bacterial Strains, Media, and Growth Condition

APP1 (serotype 1 virulent strain 4074, GenBank accession number: CP030753.1), APP5b (serotype 5b virulent strain L20, GenBank accession number: CP000569.1), and APP7 (serotype 7 virulent strain WF83, GenBank accession number: NZ_CP031869.1) were isolated and identified by our lab, and the LD50 for these strains in BALB/c mice was determined to be 5.0 × 10^6^ CFU for APP1, 5.0 × 10^7^ CFU for APP5b, and 1.0 × 10^9^ CFU for APP7. These strains were cultured on tryptic soy broth (TSB) or tryptic soy agar (TSA) with 10% calf serum and supplemented with 10 μg/mL nicotinamide adenine dinucleotide (Sangon Biotech [Shanghai] Co., Ltd., Shanghai, China) at 37 °C in a 5% CO_2_ atmosphere, 220 rpm. The galT expression strain *E. coli* BL21 (pET-galT) used in this study was transformed with the recombinant plasmid constructed in our previous study [24]. *E. coli* DH5α and *E. coli* BL21 (Beijing Tsingke Biotech Co., Ltd., Beijing, China) were grown in Luria–Bertani (LB) broth supplemented with 50 μg/mL kanamycin at 37 °C and 220 rpm.

### 4.2. Animals

Five-week-old female specific pathogen-free (SPF) BALB/c mice were purchased from Chengdu Dossy Experimental Animal Co., Ltd. (Chengdu, China), and the animal production license is No. SCXK (Chuan) 2020-0030. After a week of adaptive feeding, the mice were randomly divided into different groups. All mice were housed under standard laboratory conditions with ad libitum access to chow and water.

### 4.3. Bioinformatics Analysis for rgalT-galU

Before the commencement of the experimental procedures, computational methodologies were utilized to predict the hydrophilicity, stability, structural characteristics, and antigenic properties of rgalT-galU. Hydrophilicity analysis was conducted using the ProtScale tool (ExPASy; https://web.expasy.org/protscale/, accessed on 4 October 2024), while stability was assessed with ProtParam (ExPASy; https://web.expasy.org/protparam/, accessed on 4 October 2024). Structural predictions were performed using the SWISS-MODEL(https://swissmodel.expasy.org/, accessed on 4 October 2024), and DNASTAR Lasergene (version 17; DNASTAR Inc., Madison, WI, USA; https://www.dnastar.com) predicted antigenicity based on the Jameson–Wolf mathematical model. Furthermore, nucleotide and corresponding amino acid sequences of galT and galU from serotypes 1–18 of APP were retrieved from the NCBI database (serotype 19 data are unavailable). These sequences encompassed the following strains: S4074, S1536, S1421, M62, K17, L20, Femo, WF83, 405, CVJ13261, D13039, 56153, 1096, N273, 3906, HS143, A-8514, 16287-1, and 7311555. Conservation analysis of the galT and galU proteins among these strains was conducted using bioinformatics tools, including the R package Se-qing (version 3.3-3) from CRAN (https://cran.r-project.org/web/packages/seqinr/ (accessed on 15 October 2024)), MSAViewer (https://msa.biojs.net/ (accessed on 15 October 2024)), the Biostrings package from Bioconductor (https://bioconductor.org/packages/Biostrings (accessed on 15 October 2024)), and ggplot2 for visualization (https://cran.r-project.org/web/packages/ggplot2/ (accessed on 15 October 2024)).

### 4.4. Construction and Verification of Recombinant Plasmid pET28a-galT-galU

The recombinant plasmid pET-28a-galT was constructed and maintained by our laboratory in a previous study [24]. Primers for the galU gene were designed utilizing Oligo 7. The forward primer, galT-U-F: cactataaaaatcaaaagcttGGAGGAGGAGGAAGTATGAAAGTAATTATTCCGGTAGC, and the reverse primer, galT-U-R: gtggtggtggtggtgctcgagTTATAACGTTTTAGCTAATTTTTTA, contain underlined nucleotides denoting the addition of restriction sites (Table 1). The APP L20 (serotype 5b) strain was cultured at 37 °C overnight, after which genomic DNA was extracted using a genome extraction kit (OMEGA, Omega Bio-Tek, Norcross, GA, USA). This DNA was used as a template for the polymerase chain reaction (PCR) to amplify the galU gene. Subsequently, the pET-28a-galT plasmid was enzymes with HindIII and XhoI, and the galU fragment was ligated into this vector using a Seamless Cloning Mix (Biomed, Beijing, China). The resulting recombinant construct was transformed into *E. coli* DH5α and verified by DNA sequencing to ensure the accuracy of the insert. The pET28a-galT-galU sequencing results are shown in Figure 6.

### 4.5. Expression and Purification of rgalT-galU

The pET-galT-galU construct was cultured overnight until it reached an optical density at 600 nm of 0.6~0.8. Induction was performed using 1 mM of isopropyl-beta-D-1-thiogalactopyranoside (IPTG) (Solarbio, Beijing, China), followed by incubation at 18 °C for 24 h. After incubation, bacteria were harvested by centrifugation at 12,000× *g* for 10 min. Since rgalT-galU lacks a signal peptide and is not secreted into the culture medium, the recombinant protein was expressed intracellularly and recovered from the soluble fraction of the bacterial lysate following sonication. The resulting bacteria were suspended in 40 mL of 1× binding buffer (Bio-Rad, Hercules, CA, USA) and subjected to ultrasonication at 4 °C. After ultrasonication, the lysates were centrifuged at 12,000× *g* for 10 min to collect the soluble fraction of the cell lysate containing the His-tagged rgalT-galU. The rgalT-galU with His-tag was purified using Ni-affinity chromatography and stored at −80 °C.

### 4.6. Western Blotting

Western blotting was performed as described previously [24]. The membrane was visualized using Clarity Western ECL Substrate (Bio-Rad, Hercules, CA, USA), and pictures were taken using a ChemiDoc imaging system (Bio-Rad, Hercules, CA, USA) (BIO-RAD, USA). ImageJ2 software (version 2.16.0; NIH, USA; https://imagej.net/) was used to quantify the band intensity of the Western blot through densitometry.

### 4.7. Immunization and Challenge

To evaluate the immunogenicity of rgalT-galU, mice were immunized according to the protocol outlined in Table 1. Immunizations were administered on days 0 and 14, with no mortality observed throughout this study. Serum samples were collected via orbital vein bleeding on days 0, 8, and 15 to monitor the immune response. One week after the booster immunization, three mice from each group were selected for a splenocyte proliferation assay. Additionally, another immunization experiment was conducted to assess the protective efficacy of rgalT-galU. Furthermore, mice were immunized according to the immunization protocol outlined in Table 2. The dose of the commercial vaccine for mice was determined based on the body weight of a 4-week-old pig (approximately 10 kg) and the average body weight of the mouse (12.5 g). This resulted in a vaccine dose of 2.5 µL per mouse, which was selected to ensure proper immune stimulation in the murine model. To establish a mouse challenge model, we referenced previous studies from our laboratory, which determined the LD_50_ values for the APP1, APP5b, and APP7 strains as 5.0 × 10^6^ CFU, 5.0 × 10^7^ CFU, and 1.0 × 10^9^ CFU [52]. Survival was monitored and recorded over an 8-day post-challenge period.

Commercial vaccine is a porcine subunit *Actinobacillus pleuropneumoniae* vaccine, inactivated, PORCILIS^®^ APP (Intervet International B.V., Boxmeer, The Netherlands), which contains antigen components, including outer membrane proteins (OMPs) and Apx toxins (Apx I, Apx II, Apx III). Five-week-old mice were randomly divided into five groups (*n* = 7/group) and vaccinated subcutaneously on their backs; each group of proteins was diluted to 200 μL with PBS. The adjuvant used was MONTANIDETM Gel 02 S.T., added at a concentration of 10% (*v*/*v*) in each group.

### 4.8. Indirect Enzyme-Linked Immunosorbent Assay

Serum samples were collected from three mice per group, with each mouse representing an independent biological replicate. For each sample, three technical replicates were performed to ensure assay consistency.

To measure IgG levels, an indirect ELISA was employed. Initially, a 96-well plate (Sangon Biotech, Shanghai, China) was coated with recombinant proteins rgalT-galU, rgalT, and rgalU at a concentration of 1 μg/1001 μL, with 100 μL allocated per well. The ELISA plate was then incubated at 37 °C for 1 h, followed by a prolonged incubation at 4 °C for 10–12 h. After incubation, the liquid in each well was discarded, and a PBST wash solution was added at a volume of 220 μL per well. The plate was subjected to shaking and washing at 300 rpm for 3 min, a process repeated three times to ensure thorough cleaning. Subsequently, 5% skim milk was introduced into each well at 100 μL per well, and the plate was incubated at 37 °C for 1.5 h to block non-specific binding sites. The washing procedure with PBST was repeated three times. Mouse serum, collected post-immunization, was serially diluted with 5% skim milk (configured with PBST) and added to the wells pre-coated with the antigens, at a volume of 100 μL per well. Each dilution was tested in triplicate, and a negative control was included for comparison. The ELISA plate was incubated at 37 °C for 1 h. Following incubation, 220 µL of PBST wash solution was added to each well, and the wells were washed three times. Subsequently, each well was incubated with goat anti-mouse HRP-conjugated IgG, diluted at a ratio of 1:5000, at 37 °C (100 µL per well) (Boster Biological, Wuhan, China) for 30 min. After three additional washes with PBST, 100 µL per well of TMB chromogenic reagent (Solarbio, Beijing, China) was added to the 96-well ELISA plate, which was kept in a shaded area and allowed to react for 15 min at room temperature. The reaction was terminated by adding 50 µL of 2 M sulfuric acid to each well to quench the enzyme activity, and the optical density (OD) value was measured at 450 nm.

To ascertain the IgG titers, a modified version of the indirect ELISA assay method, as outlined by Chen et al. [53], was employed. In this procedure, 100 µL of inactivated bacterial solutions of APP5b, APP1516, and APP2701, each with a concentration of 1.665 × 10^7^ CFU/mL, was dispensed into individual wells of a 96-well ELISA plate (Sangon Biotech, Shanghai, China) for coating purposes. The plate was incubated at 37 °C until the solutions were completely dried. Subsequently, 5% skim milk in PBST (100 µL per well) was added to each well and incubated at 37 °C for 1 h to block. Following the blocking step, each well underwent three washes with 220 µL of PBST wash solution. The subsequent steps, including primary antibody incubation, washing, secondary antibody incubation, washing, color development, reaction termination, and OD value determination, were conducted according to the methods previously described for IgG level measurement. A test well was deemed positive if its OD450 value was ≥0.1 and the P/N ratio (the ratio of the OD450 value of the test well to that of the negative control well) was ≥2.1. The highest dilution of serum that tested positive was designated as the antibody titer of the serum.

### 4.9. Splenocyte Proliferation Assay

Splenocytes were isolated aseptically with a mouse lymphocyte isolation kit (Solarbio, Beijing, China) following the manufacturer’s instructions. The splenocyte proliferation assay was prepared as previously described [24]. After stimulation with rgalT-galU, rgalT, and rgalU (10 μg/well) and separately with ConA (Sigma, St. Louis, MO, USA), splenocytes were cultured for 48 h at 37 °C. Cell proliferation was evaluated using the CCK8 Cell Proliferation Assay Kit (Beyotime, Shanghai, China) according to the manufacturer’s protocol, and the splenocyte proliferation rate was subsequently calculated. Spleens were obtained from three mice per group, with each mouse serving as an independent biological replicate. For each sample, three technical replicates were conducted to assess reproducibility.

### 4.10. Assessment of Cytokines in Serum

According to the manufacturer’s protocols, the IFN-γ, IL-12, and IL-4 levels in serum samples were quantified using a double antibody mouse sandwich ELISA kit (Elabscience, Wuhan, China). These cytokines were selected as markers to evaluate immunogenic efficacy due to their roles in modulating immune responses: IFN-γ and IL-12 are key indicators of Th1-type cell-mediated immunity, while IL-4 reflects Th2-type humoral immunity. Absorbance was measured at wavelengths of 450 nm and 630 nm. Serum samples were collected from three mice per group, with each mouse representing an independent biological replicate. For each sample, three technical replicates were performed to ensure assay consistency.

### 4.11. Histopathological Analysis

Mouse lung tissue was aseptically dissected, rinsed with PBS, and fixed in 4% paraformaldehyde. The fixed tissues were paraffin-embedded, stained with hematoxylin and eosin (H&E), and examined by optical microscopy. Lung tissue damage was evaluated using a standardized 4-point scale (0 = normal, 4 = severe) assessing four parameters: (1) lesion extent, (2) inflammatory infiltration, (3) fibrosis, and (4) tissue damage (hemorrhage/necrosis). The lung tissue damage score was calculated as (Extent + 2 × Inflammation + Fibrosis + 2 × Damage)/6, with three biological replicates (mice) analyzed per group [54,55].

### 4.12. Immunohistochemical Analysis

A previously reported method was employed for immunohistochemistry (IHC) analysis [24]. A rabbit anti-mouse myeloperoxidase (MPO) monoclonal antibody was used for staining to assess neutrophil infiltration in lung tissues. Images of the stained sections were taken at 400× magnification with Image J2 (version 6.0, USA). Each image’s integrated optical density (IOD) served as an indicator of positivity. Lung tissues were obtained from three mice per group, with each mouse serving as an independent biological replicate.

### 4.13. Statistical Analysis

Data are presented as mean ± standard deviation (SD). Statistical analyses were performed to evaluate both within-group and between-group differences. For within-group comparisons, paired two-tailed *t*-tests were used. For between-group comparisons, one-way analysis of variance (ANOVA) was performed using GraphPad Prism 10.0 (GraphPad Software, San Diego, CA, USA; https://www.graphpad.com/). For multiple comparisons, Tukey’s post hoc test was applied to control type I errors. Statistical significance was determined at the levels of * *p* < 0.05, ** *p* < 0.01, and *** *p* < 0.001, with all adjusted *p*-values indicated in the figures and text where applicable.

## 5. Conclusions

In conclusion, our study demonstrates that the rgalT-galU is a highly promising candidate for developing a broad-spectrum subunit vaccine against PCP. The protein elicits a potent immune response characterized by high IgG titers, robust lymphocyte proliferation, and increased cytokine production, leading to significant protection against multiple APP serotypes. The reduction in pulmonary lesions and neutrophil infiltration further highlights the therapeutic potential of rgalT-galU. These findings provide a strong foundation for future research to optimize and commercialize this vaccine candidate, aiming to reduce the economic and health burdens associated with PCP in the global swine industry.

## Figures and Tables

**Figure 1 ijms-26-03634-f001:**
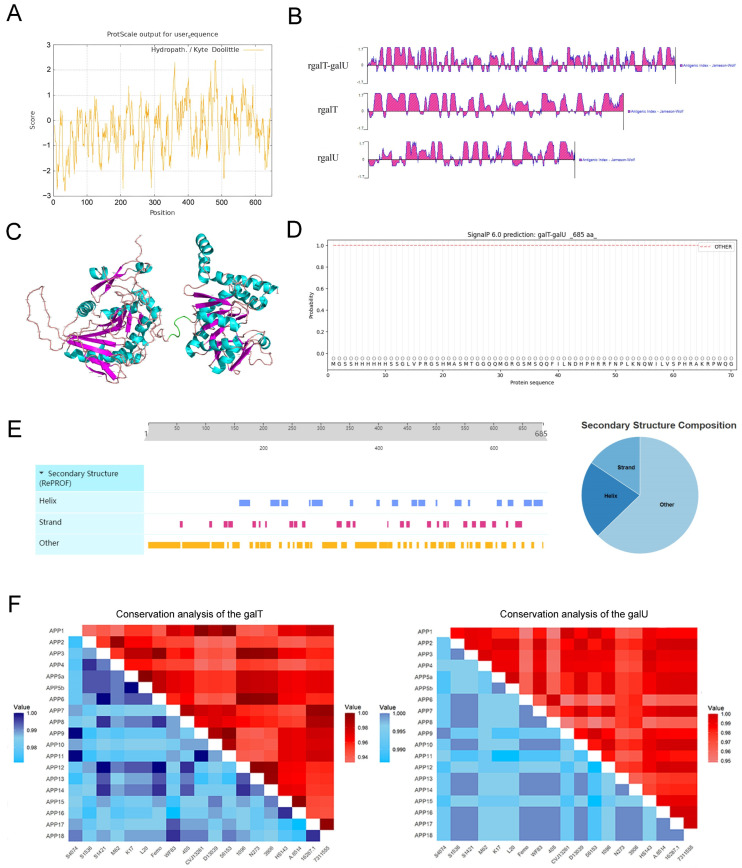
Bioinformatics analysis of rgalT-galU. (**A**) Hydrophilicity analysis using ProtScale (computational prediction). (**B**) Antigenicity prediction based on computational modeling using the Jameson–Wolf mathematical model. (**C**) Structural prediction using SWISS-MODEL (in silico modeling). α-helixes are shown in cyan, β-sheets are in purple, random-coil is in pink, and linker (GGGGS) is in green. (**D**) Signal peptide analysis using SignalP-6.0 (computational prediction). (**E**) Structural elements analysis. (**F**) Heatmap showing conservation analysis of galT and galU based on nucleotide and amino acid sequence alignments. Red represents nucleotide sequence conservation, and blue represents amino acid sequence conservation.

**Figure 2 ijms-26-03634-f002:**
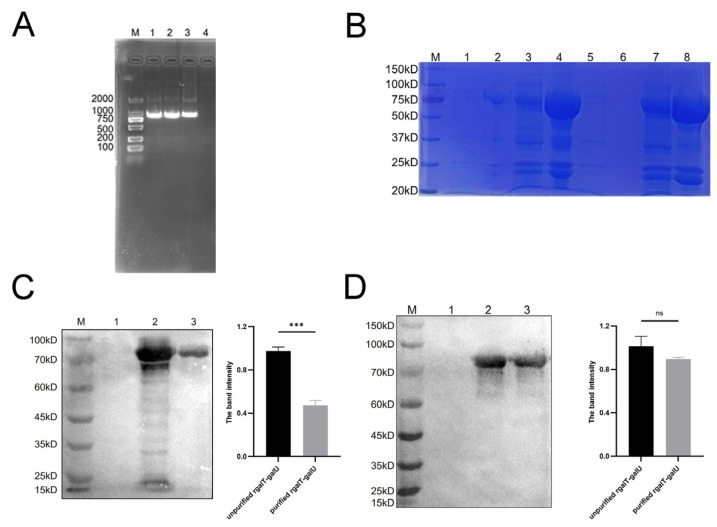
Expression and identification of rgalT-galU. (**A**) Amplified galU fragment. Line 1–3: galU fragment. Line 4: Negative control. M: Trans2K^®^ DNA Marker (ABclonal Biotechnology Co., Ltd., Wuhan, China). (**B**) SDS-PAGE analysis of rgalT-galU. M: PageRulerTM Prestained Protein Ladder. Lines 1 and 5: pET28a BL21 (DE3) control. Lines 2 and 6: pET28a-galT-galU BL21 (DE3) without IPTG. Lines 3 and 7: pET28a-galT-galU BL21 (DE3) inclusion body. Lines 4 and 8: pET28a-galT-galU BL21 (DE3) soluble fraction of the cell lysate. (**C**) Western blotting analysis of galT-galU used His-tag monoclonal antibody. M: PageRulerTM Prestained Protein Ladder. Line 1: pET28a BL21 (DE3) control. Line 2: Unpurified rgalT-galU. Line 3: Purified rgalT-galU. (**D**) Western blotting analysis of galT-galU used galT-galU polyclonal antibody. M: PageRulerTM Prestained Protein Ladder. Line 1: pET28a BL21 (DE3) control. Line 2: Unpurified rgalT-galU. Line 3: Purified rgalT-galU. ns indicates no significant difference from the control group (*p* > 0.05); *** *p* < 0.001.

**Figure 3 ijms-26-03634-f003:**
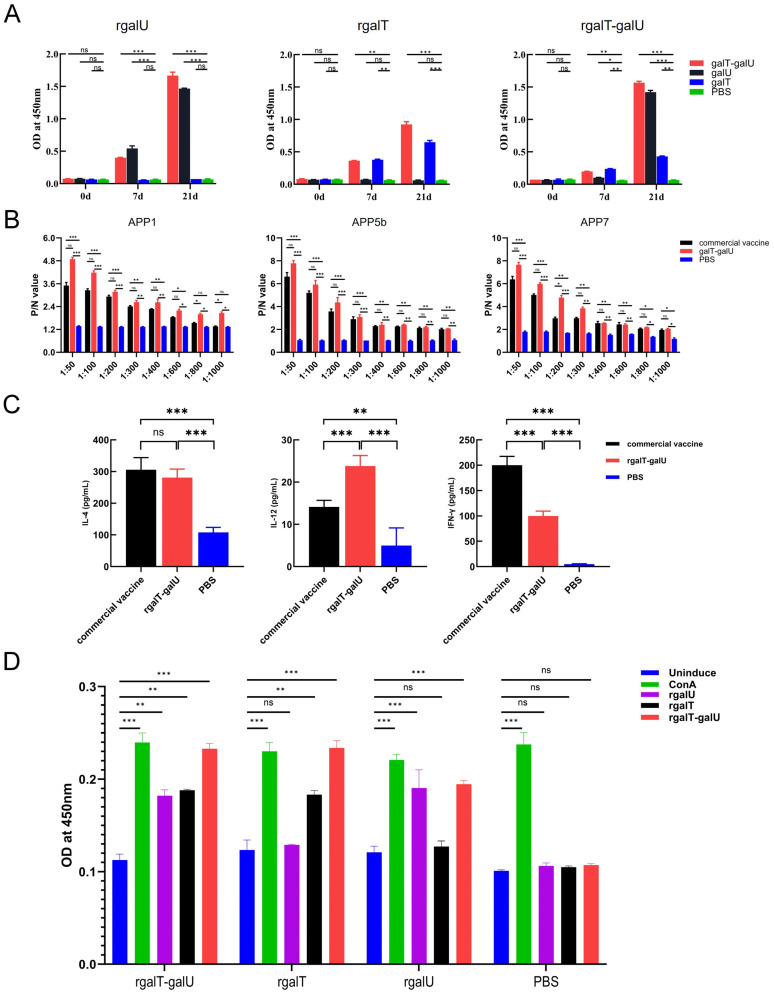
Immunogenicity evaluation of rgalT-galU. (**A**) Determination of serum IgG level. The IgG level was assessed by an indirect ELISA coated with rgalT, rgalU, and rgalT-galU, respectively. IgG levels were shown as OD values at 450 nm. (**B**) Determination of serum IgG titer. The IgG titer was assessed by an indirect ELISA coated with APP1, APP5b, and APP7, respectively. IgG titers were shown as P/N values. (**C**) Detection of IFN-γ, IL-4, and IL-12 in mouse serum. The absorbance was measured at 450 and 570 nm to calculate the cytokine concentration. (**D**) The splenocyte proliferation assay (*n* = 3). Levels of proliferation were detected using the CCK8 method, and the results were absorbance at 450 nm. Data are represented as mean ± SD. ns indicates no significant difference from the control group (*p* > 0.05); * *p* < 0.05, ** *p* < 0.01, *** *p* < 0.001.

**Figure 4 ijms-26-03634-f004:**
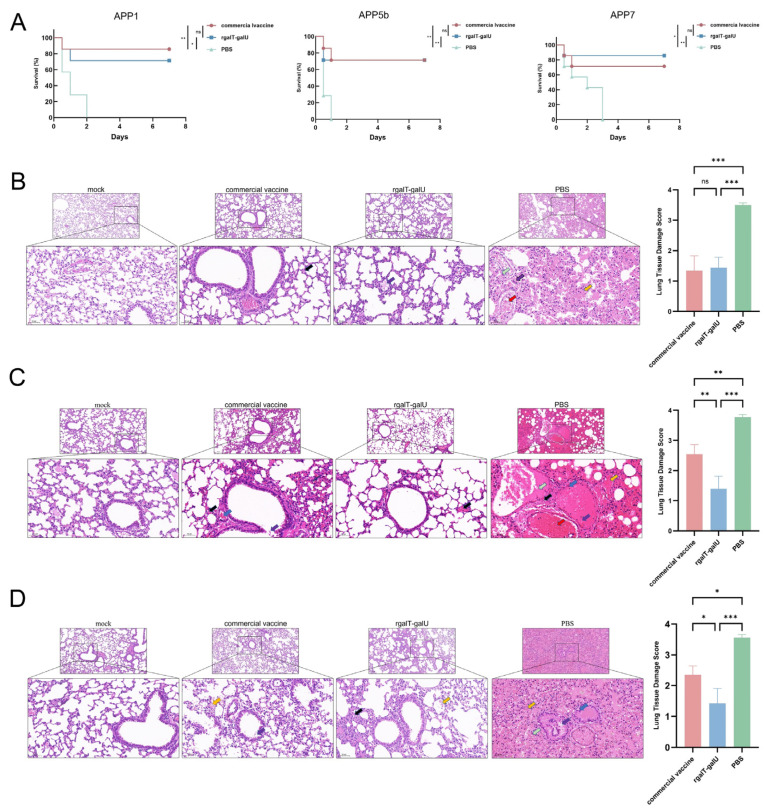
Animal survival rates and lung histopathological analysis in mice. (**A**) Survival rates of mice within 7 days following challenge with APP1, APP5b, and APP7, respectively (*n* = 7 mice per group). (**B**–**D**) Pathological observations of lung tissue in mice following challenges with APP1, APP5b, and APP7, respectively. Smaller images are shown at a lower magnification (100×), while larger images are shown at a higher magnification (200×). Black arrows indicate inflammatory cell infiltration, blue arrows indicate necrotic and detached epithelial cells, purple arrows indicate eosinophilic substances, green arrows indicate tissue exudates, yellow arrows indicate hemorrhage, and red arrows indicate congestion. Lung tissue damage scores (shown in Figure 4B–D) were derived from these parameters using the standardized scoring system described in the Methods Section. Data are represented as mean ± SD. *n* = 3. ns indicates no significant difference from the control group (*p* > 0.05); * *p* < 0.05, ** *p* < 0.01, *** *p* < 0.001.

**Figure 5 ijms-26-03634-f005:**
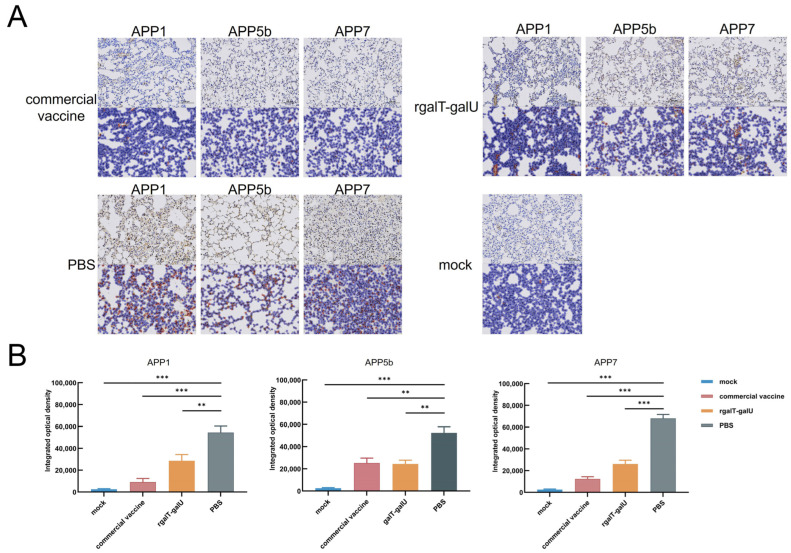
Immunohistochemical analysis of neutrophil infiltration in mouse lungs. (**A**) Neutrophil infiltration in the lungs of mice from each group was analyzed by IHC at 100× magnification, with representative results shown for each group. The MOCK group, consisting of non-infected healthy mice, served as the baseline control in our study. MPO-positive neutrophils are highlighted in red using Aipathwell software (Version 2; Servicebio, China; https://www.servicebio.com) for better visibility. The mock group served as the negative control. (**B**) IOD values of lung neutrophils for each group were analyzed using ImageJ2 software (Version 2.16.0; NIH, USA; https://imagej.net/). Data are represented as mean ± SD. *n* = 3. ** *p* < 0.01; *** *p* < 0.001.

**Figure 6 ijms-26-03634-f006:**
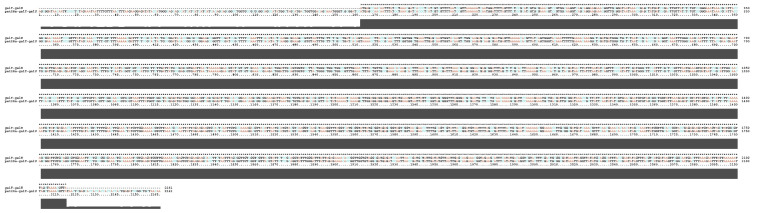
The sequencing results of pET28a-galT-galU. The asterisks (*) represent conserved positions across the aligned sequences, where the nucleotides (or amino acids) are identical in all samples.

**Table 1 ijms-26-03634-t001:** PCR primer sequences used in this study.

Primer Name	Primer Sequence (5′–3′)	Size	Enzyme	Fragment Length
galT-U-F	cactataaaaatcaaaagcttGGAGGAGGAGGAAGT ATGAAAGTAATTATTCCGGTAGC	59 bp	Hind III	888 bp
galT-U-R	gtggtggtggtggtgctcgagTTATAACGTTTTAGCT AATTTTTTA	46 bp	Xho I	

Note: The underlined sites are the restriction sites.

**Table 2 ijms-26-03634-t002:** The immunization protocol in mice.

Group	Boost Immunization Dose (0 Day)	Secondary Immunization Dose (14 Day)
Commercial vaccine	2.5 μL (12.5 g)	3.3 μL (16.5 g)
galT-galU	100 μg/200 μL	120 μg/200 μL
galT	100 μg/200 μL	120 μg/200 μL
galU	100 μg/200 μL	120 μg/200 μL
PBS	200 μL	200 μL

## Data Availability

The datasets analyzed and materials used during the current study are available from the corresponding authors upon reasonable request.

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
