# Peer review of "Development and Characterization of a Recombinant galT-galU Protein for Broad-Spectrum Immunoprotection Against Porcine Contagious Pleuropneumonia"

_ijms, 2025, doi:10.3390/ijms26083634_

Round 1

Reviewer 1 Report

Comments and Suggestions for Authors

In this manuscript the authors present the development and characterization of a recombinant galT-galU protein as a potential broad-spectrum immunoprotection against Actinobacillus pleuropneumoniae (APP). The study is well-structured and provides a comprehensive analysis, including bioinformatics, protein expression, immunogenicity evaluation, and in vivo protective efficacy. The data support the claims of cross-protection, and the findings contribute to vaccine development strategies against APP. However, some aspects require clarification, including methodology descriptions, statistical analyses, and interpretation of results.

Specifically, the manuscript presents various statistical comparisons, but it is unclear whether multiple comparisons were corrected. Please clarify the statistical approach used. Additionally, the methodology section should provide more detail on replication of experiments, such as the number of biological replicates per assay and whether experiments were repeated independently. The manuscript suggests that rgalT-galU is comparable or superior to commercial vaccines. However, additional discussion on the limitations of the commercial vaccine used for comparison or further experimental data would strengthen the argument. Finally, some figure legends lack essential details about the statistical significance and units of measurement.

Below I provide specific comments with line numbers:

Line 29-30: "Previous studies identified galT and galU as promising antigen candidates." Would be good that the authors include citations in the introduction and specify what sections in the manuscript include this information.

Line 33-34: "...demonstrated strong immunogenicity in mice, with increased IgG titers..." Specify how immunogenicity was assessed (e.g., ELISA, neutralization assays).

Line 67-68: "Currently, vaccination is still frequently utilized as a strategy to prevent the disease." It would be helpful to provide current vaccine efficacy data or a reference for context. Also consider that different regions could have slightly different strategies.

Line 130: "These dates? provide a reliable basis for expressing the rgalT-galU." Likely meant "data"

Line 185-188: "These results indicate that rgalT-galU effectively enhances both humoral and cellular immune responses..." Consider discussing the relevance of these cytokines in APP protection.

Line 297-298: "This finding suggests that rgalT-galU confers protective effects that transcend specific serotypes..." Additional discussion on potential mechanisms of cross-protection would strengthen this point. The authors could provide examples.

Line 329-330: "Integrating rgalT-galU into existing vaccine formulations..." Consider discussing whether adjuvants were tested or if future studies will evaluate their effects.

Figure Comments:

  • Figure 1: The legend should clarify whether antigenicity prediction was based on computational modeling or experimental validation.
  • Figure 2: Western blot bands should be quantified using densitometry to strengthen the results.
  • Figure 4: Indicate whether histopathological scores were used to quantify lung lesions.
  • Figure 5: Immunohistochemistry (IHC) images should include an inset showing an unstained control for reference.

Author Response

Dear editors and reviewers,

Thank you for your comments. We have read and revised the manuscript carefully to meet the demands of your wish. We hope that you will positively respond to our revision. The responses to the reviewers' comments are provided below.

Comments and Suggestions for Authors

In this manuscript the authors present the development and characterization of a recombinant galT-galU protein as a potential broad-spectrum immunoprotection against Actinobacillus pleuropneumoniae (APP). The study is well-structured and provides a comprehensive analysis, including bioinformatics, protein expression, immunogenicity evaluation, and in vivo protective efficacy. The data support the claims of cross-protection, and the findings contribute to vaccine development strategies against APP. However, some aspects require clarification, including methodology descriptions, statistical analyses, and interpretation of results. Specifically, the manuscript presents various statistical comparisons, but it is unclear whether multiple comparisons were corrected. Please clarify the statistical approach used. Additionally, the methodology section should provide more detail on replication of experiments, such as the number of biological replicates per assay and whether experiments were repeated independently. The manuscript suggests that rgalT-galU is comparable or superior to commercial vaccines. However, additional discussion on the limitations of the commercial vaccine used for comparison or further experimental data would strengthen the argument. Finally, some figure legends lack essential details about the statistical significance and units of measurement.

Response: We would like to thank you sincerely for your thoughtful and constructive feedback. We have carefully considered your suggestions and have revised the manuscript accordingly.   Below, we detail our responses to each of your comments:

  1. Statistical Comparisons and Multiple Comparisons Correction:

We have revised the Methods section to clarify our statistical approach. Specifically, we have detailed the statistical tests applied to the data and indicated whether multiple comparisons were corrected. Please refer to the revised manuscript, lines 686-693, for these updates.

  1. Replication and Experimental Details:

In response to your comment regarding experimental replication, we have expanded the Methods section to include detailed information on the number of biological replicates per assay.  

We now clearly state that each experiment was performed using its respective independent biological replicates, and that each assay was repeated independently on its respective separate occasions. This information is intended to enhance reproducibility and provide a clearer understanding of our experimental design. For more details, please refer to the revised manuscript in lines 593-595, 654- 656, 663-666, 668-674, and 683-684.

  1. Comparison with Commercial Vaccines:

We acknowledge the need for further discussion on the limitations of the commercial vaccine used for comparison. We have expanded the background on commercially available vaccines in the revised manuscript. Please refer to the revised manuscript, lines 75-93, for these updates. Additionally, we have provided a more detailed explanation of the commercial vaccine used in the study. Please refer to the revised manuscript, lines 588-589, for this clarification. Furthermore, in the results section, we have strengthened the comparison between the recombinant rgalT-galU vaccine and the commercial vaccine, highlighting the consistency of the immune response induced by rgalT-galU in contrast. Please refer to the revised manuscript, lines 260-267, for these revisions.

  1. Figure Legends Statistical Significance and Units:

We have revised the figure legends throughout the manuscript to include all necessary details regarding the statistical significance indicators (e.g., P values) and the units of measurement for the reported data. These revisions should now provide readers with all essential information to interpret the figures accurately. Please refer to the revised manuscript, lines 294-303 (Figure 3), lines 329-340 (Figure 4), lines 353-359 (Figure 5).

Below I provide specific comments with line numbers:

Line 29-30: "Previous studies identified galT and galU as promising antigen candidates." Would be good that the authors include citations in the introduction and specify what sections in the manuscript include this information.

Response: Thank you for your helpful suggestion. We have now incorporated additional citations in the Introduction to support the statement “Previous studies identified galT and galU as promising antigen candidates.” Specifically, we have included references [22] & [23] that provide detailed evidence on the promising roles of these genes as antigen candidates. Additionally, we have provided a more detailed description of the background, evaluated previous studies and highlighted existing gaps in the field. Please refer to the revised manuscript, lines 121-140, for these updates. We believe these modifications enhance the clarity and depth of our manuscript. Thank you again for your constructive feedback.

References:

[23] Zhang, F.; Zhang, Y.; Wen, X.; Huang, X.; Wen, Y.; Wu, R.; Yan, Q.; Huang, Y.; Ma, X.; Zhao, Q.; Cao, S., Identification of Actinobacillus pleuropneumoniae Genes Preferentially Expressed During Infection Using In Vivo-Induced Antigen Technology (IVIAT). J. Microbiol. Biotechnol. 2015, 25, (10), 1606-13.

[24] Zhang, F.; Cao, S.; Zhu, Z.; Yang, Y.; Wen, X.; Chang, Y. F.; Huang, X.; Wu, R.; Wen, Y.; Yan, Q.; Huang, Y.; Ma, X.; Zhao, Q., Immunoprotective Efficacy of Six In vivo-Induced Antigens against Actinobacillus pleuropneumoniae as Potential Vaccine Candidates in Murine Model. Front Microbiol 2016, 7, 1623.

Lines 33-34: "...demonstrated strong immunogenicity in mice, with increased IgG titers..." Specify how immunogenicity was assessed (e.g., ELISA, neutralization assays).

Response: We sincerely appreciate your insightful feedback. In response to your suggestion, we have revised the Abstract and Method sections to provide a clearer explanation of the immunogenicity assessment. Specifically, we now detail that the immunogenicity of rgalT-galU was evaluated by measuring IgG levels and IgG titers using indirect ELISA. Furthermore, we have expanded on the immune response assessment by including information on splenic lymphocyte proliferation and cytokine assays. For the updated Abstract, please refer to lines 34-44 of the revised manuscript. Detailed procedures for assaying IgG levels and titers are provided in section 4.9, "Indirect Enzyme-Linked Immunosorbent Assay," on lines 593-645. Additionally, this information has been incorporated into the legend of Figure 3, as indicated on lines 294-303.

Line 67-68: "Currently, vaccination is still frequently utilized as a strategy to prevent the disease." It would be helpful to provide current vaccine efficacy data or a reference for context. Also consider that different regions could have slightly different strategies.

Response: We appreciate your insightful feedback. In response to your suggestion, we have meticulously revised the manuscript to incorporate pertinent vaccine efficacy data, along with contextual references. The updated version addresses the clinical limitations of commercially available vaccines, specifically their serotype-specific protection and the challenges associated with achieving cross-protection. Additionally, we have included information on the diverse strategies implemented across various regions, with specific examples of vaccines such as Aptovac, Serkel PleuroAP, and Neumosun. While these vaccines provide partial protection, they encounter limitations in cross-serotype protection. Furthermore, we have emphasized the critical need for the development of novel genetically engineered subunit vaccines, given the substantial economic impact of APP on the swine industry. Please consult lines 79-91 of the revised manuscript for these updates.

Line 130: "These dates? provide a reliable basis for expressing the rgalT-galU." Likely meant "data".

Response: We express our sincere gratitude to the reviewer for highlighting this issue, and we apologize for the oversight. We have amended the wording to "data" in the revised manuscript, which can be found on line 189. We appreciate your valuable feedback.

Line 185-188: "These results indicate that rgalT-galU effectively enhances both humoral and cellular immune responses..." Consider discussing the relevance of these cytokines in APP protection.

Response: We express our gratitude for your insightful suggestion. In response, we have amended the manuscript to incorporate a discussion on the significance of these cytokines in the context of APP protection. In particular, we have expanded our analysis of the roles of IFN-γ, IL-4, and IL-12 in orchestrating immune responses against APP. The revised discussion is located on lines 401-424 of the manuscript.

Line 297-298: "This finding suggests that rgalT-galU confers protective effects that transcend specific serotypes..." Additional discussion on potential mechanisms of cross-protection would strengthen this point. The authors could provide examples.

Response: Thank you for your valuable suggestion. We have expanded the discussion on the potential mechanisms of cross-protection in the manuscript. Please refer to the revised manuscript, lines 428-446, for these updates.

Line 329-330: "Integrating rgalT-galU into existing vaccine formulations..." Consider discussing whether adjuvants were tested or if future studies will evaluate their effects.

Response: Thank you for your valuable feedback. We regret not including information on adjuvants in the initial manuscript. In the updated version, we have incorporated relevant details about adjuvants on lines 590-591. Furthermore, we have explained in the discussion that while adjuvants were not examined in this study, future research using swine models will involve screening and evaluating adjuvants to enhance vaccine effectiveness. Please see lines 476-485 in the revised manuscript for these changes.

Figure Comments:

Figure 1: The legend should clarify whether antigenicity prediction was based on computational modeling or experimental validation.

Response: We appreciate your suggestion and have accordingly revised the figure legend to explicitly indicate that all results depicted in Figure 1, encompassing hydrophilicity analysis, antigenicity prediction, structural prediction, and signal peptide analysis, are derived from computational modeling. For further details, please consult the updated manuscript on lines 207-216.

Figure 2: Western blot bands should be quantified using densitometry to strengthen the results.

Response: We sincerely appreciate your insightful suggestion. In response, we have utilized ImageJ software to quantify the density of protein bands in the Western blot analysis, thereby improving the interpretation of the results. Please refer to the updated Figures 2C and 2D, their accompanying legends, and lines 237-248 for the revisions. Furthermore, the methodology section about Western Blotting has been revised; please refer to lines 566-567 for the updated content.

Figure 4: Indicate whether histopathological scores were used to quantify lung lesions.

Response: We appreciate your insightful feedback. In response, we have incorporated histopathological scores to assess the severity of lung lesions quantitatively. Consequently, Figure 4 has been revised, and the figure legend has been updated to reflect these modifications. We invite you to review the revised Figures 4B, 4C, and 4D, along with their updated legends, which are located on lines 328-340. Furthermore, a comprehensive description of the histopathological scoring methodology is now included in the Methods section. Please consult lines 668-674 in the revised manuscript for these updates. The scoring methodology for histopathological analysis was derived from the following literature sources:

[53]. Gustavo Matute-Bello, G.D., Bethany B Moore, Steve D Groshong, Michael A Matthay, Arthur S Slutsky, Wolfgang M Kuebler An official American Thoracic Society workshop report: features and measurements of experimental acute lung injury in animals. Am J Respir Cell Mol Biol, 2011. 44(5): p. 725-738.

[54]. Kolb, J.G.M., Animal models of pulmonary fibrosis: how far from effective reality? Am J Physiol Lung Cell Mol Physiol, 2008. 294(2): p. L151.

Figure 5: Immunohistochemistry (IHC) images should include an inset showing an unstained control for reference.

Response: Thank you for your insightful feedback regarding the inclusion of an unstained control in our immunohistochemistry (IHC) images. While we did not incorporate an unstained control in our experiments, we employed a MOCK group consisting of non-infected, healthy mice that were not exposed to the bacteria, serving as our baseline control. This MOCK group underwent identical tissue processing, staining protocols, and imaging conditions as the experimental groups, effectively accounting for any nonspecific binding or background staining in the absence of the pathogen.​

To further validate our findings, we meticulously examined adjacent normal tissue structures within each section, consistently observing negligible background staining. This consistency reinforces the specificity of our IHC results.​

In response to your suggestion and to enhance the clarity of the myeloperoxidase (MPO) staining, we have updated the images using Aipathwell software to highlight MPO-positive neutrophils in red, improving their visibility. Please refer to the revised Figure 5A and its accompanying legend on lines 355–359 for further details.​

The absence of observable nonspecific staining or background signal in the MOCK control tissues, as shown in Figure 5A, supports the specificity of the IHC results in our experimental groups. Nonetheless, we acknowledge that including an unstained control would further strengthen the rigor of our methodology and will incorporate this valuable suggestion in future studies to ensure comprehensive validation of our staining protocols.​

Please let us know if additional clarifications or data are required.

Reviewer 2 Report

Comments and Suggestions for Authors

  1. In the introduction section please specify, what gaps remained to be evaluated after you conducted the two previous [20, 21] studies – this will help readers to understand the goals of the current study.
  2. Table 1 – I recommend using a heat-map. Besides, since all the serotypes are conservative within these two regions, the table can be moved to supplementary data.
  3. While the similarity of galT and galU was discussed, the rest of the bioinformatic analysis has not been discussed. What conclusions did authors draw based on the data presented (Fig1)?
  4. Were all the parameters presented in Fig 1 compared between galT, galU and galT galU?
  5. Figure 2: in the Western blot analysis there are too many bands (considering that the purified antigen was used). I recommend optimizing the conditions.
  6. Showing the antigen before and after purification will be beneficial for the manuscript.
  7. Since the target protein secreted in the supernatant (at higher concentrations compare to lysate) it remains unclear, why for the vaccine composition the lysate was used.
  8. How did the authors conclude that 2.5 µl of commercial vaccine would be an appropriate amount to reflect the natural conditions (when used for pigs)?
  9. Did authors test whether serum samples from commercial vaccine vaccinated animals reacted with the recombinant antigens (Fig. 3A)? What were the titers?
  10. In the introduction the limitations of commercial vaccines were mentioned in the aspect of cross-protection. However, the data in Fig 3B shows that the commercial vaccine could induce a strong immune response against all the serotypes used in this study.
  11. 9-1 what is the commercial vaccine serotype?
  12. I recommend analyzing the antibody titers for one antigen/vaccine between different serotypes (APP1, APP5b, APP7). This analysis may help to identify that new vaccine can induce strong immune response which does not differ between the strain while the commercial vaccine tends to induce stronger immune response to one strain but for the strain 2 it is significantly lower ….

Author Response

Dear editors and reviewers,

Thank you for your comments. We have read and revised the manuscript carefully to meet the demands of your wish. We hope that you will positively respond to our revision. The responses to the reviewers' comments are provided below.

Comments and Suggestions for Authors:

1.In the introduction section please specify, what gaps remained to be evaluated after you conducted the two previous [20, 21] studies – this will help readers to understand the goals of the current study.

Response: We are grateful for your insightful suggestion. In response, we have enriched the introduction section by undertaking a comprehensive literature review and synthesizing key findings. This process has enabled us to clearly identify distinct research gaps that warrant further investigation, as outlined below.

In pigs that have survived either natural or experimental infections with APP, the host's internal environment—characterized by factors such as limitations in NAD or iron—can induce the bacterial expression of certain conserved antigens, including Apx toxin and TbpB protein. These antigens, which may be conserved across various serotypes, potentially contribute to cross-serotype immunoprotection [21,22]. However, achieving robust cross-serotype protection likely necessitates synergistic interactions among multiple antigens, highlighting the need for further validation of their expression mechanisms. Existing studies predominantly focus on specific APP serotypes (e.g., serotype 10) and controlled growth conditions, yet they lack data on cross-serotype protection, which is crucial for practical field applications. Additional research gaps have been identified as follows: First, there is a deficiency in long-term immunity assessment. The duration of protection, the necessity for booster vaccinations, and the rate of immune decline have yet to be evaluated. Second, there is a lack of comprehensive safety evaluations. Current data is insufficient to adequately assess adverse effects or impacts on pig health and growth. Third, there is a need for validation of field applicability. The efficacy of vaccines under real-world conditions, including co-infections, stress, and farm management practices, has not been confirmed. Future research should prioritize these areas to advance the development of broadly protective vaccines against APP.

Additionally, the revised manuscript underscores that while galT demonstrates significant protective effects against APP5b and galU exhibits partial protective effects alongside superior immune-stimulatory properties, the cross-serotype protective efficacy of these antigens remains inadequately assessed. Furthermore, the potential synergistic effect of the galT-galU fusion protein has yet to be explored.

For detailed updates, please refer to lines 121-151 of the revised manuscript.

2.Table 1 – I recommend using a heat-map. Besides, since all the serotypes are conservative within these two regions, the table can be moved to supplementary data.

Response: We appreciate your valuable suggestion. In response, we have integrated your recommendation by substituting the initial conservation analysis table with heat maps that depict sequence similarity across various serotypes for both the galT and galU genes. The original tables have been relocated to the supplementary data section to preserve the detailed numerical values. Kindly refer to the updated Figure 1F and Supplemental Table 1 for these modifications.

3.While the similarity of galT and galU was discussed, the rest of the bioinformatic analysis has not been discussed. What conclusions did authors draw based on the data presented (Fig1)?

Response: We would like to express our sincere gratitude to the reviewer for their valuable feedback. In response, we have enhanced the Results section of the revised manuscript by including an in-depth discussion of the bioinformatic analyses depicted in Figure 1. Additionally, we propose that the galT-galU fusion protein holds considerable promise as a candidate molecule for cross-serotype vaccine development, due to its advantageous physicochemical properties, structural benefits, and conservation of the galT-galU gene. For additional information, please refer to lines 171-183 and 191-204 of the revised manuscript.

4.Were all the parameters presented in Fig 1 compared between galT, galU and galT galU?

Response: We acknowledge the reviewer's insightful query. In Figure 1B, which illustrates the antigenic index, we conducted a direct comparison among galT, galU, and the rgalT-galU fusion protein. In contrast, the remaining panels—hydropathy, signal peptide, and secondary structure—were exclusively focused on the fusion protein. This approach was intentional, as our primary objective was to assess the immunological and physicochemical properties of the fusion protein. We have provided clarification on this matter in the updated figure legend and manuscript text. For further details, please refer to lines 171-183 of the revised manuscript.

5.Figure 2: in the Western blot analysis there are too many bands (considering that the purified antigen was used). I recommend optimizing the conditions.

Response: We express our gratitude to the reviewer for their insightful feedback. In response, we have refined the Western blot conditions by employing a novel His-tag monoclonal antibody in conjunction with a galT-galU polyclonal antibody. This adjustment has yielded clearer results, devoid of extraneous bands. Consequently, Figure 2 has been updated to reflect these improvements. We invite you to examine Figures 2C and 2D, along with the accompanying legend (lines 243–248). We appreciate your valuable suggestion.

6.Showing the antigen before and after purification will be beneficial for the manuscript.

Response: We express our gratitude for the reviewer's insightful suggestion. In response, we have incorporated the antigen prior to purification in the Western blot analysis. Please refer to Figures 2C and 2D, along with the corresponding legend (lines 243248), for further details. We appreciate your valuable feedback.

7.Since the target protein secreted in the supernatant (at higher concentrations compare to lysate) it remains unclear, why for the vaccine composition the lysate was used.

Response: We appreciate the reviewer’s comment and would like to clarify that rgalT-galU was not secreted into the culture medium. As predicted by SignalP 6.0 (Figure 1D), the protein lacks a signal peptide and is thus expressed intracellularly. In our previous manuscript, the term "supernatant" referred to the soluble fraction of the bacterial lysate following cell disruption via sonication, not the culture supernatant. We have revised the manuscript to use more precise terminology throughout, ensuring clarity and avoiding any misunderstandings. Please refer to the revised manuscript in line 242. For the revised methods section, please see lines 555-557 and 560.

8.How did the authors conclude that 2.5 µl of commercial vaccine would be an appropriate amount to reflect the natural conditions (when used for pigs)?

Response: We acknowledge the reviewer's insightful comment. The commercial vaccine Porcilis® APP contains 600 mg of antigen concentrate per 2 mL dose; however, the precise weight of the purified protein components (such as Apx toxins and outer membrane proteins) remains unspecified, as these components are quantified in arbitrary units (50 units per antigen). In the absence of molecular weight data for these units, we were unable to calculate the protein content by weight directly. Although we considered employing the allometric scaling method to determine the appropriate mouse dose, this approach resulted in a dose of 13.33 mL, which is impractical. Consequently, we adopted a body-weight-based dose conversion approach to determine an appropriate dose. We calculated the mouse dose based on the body weight of a 4-week-old pig (approximately 10 kg) and the average body weight of a mouse (12.5 g). This conversion yielded a vaccine dose of 2.5 µL per mouse. This dose was selected to maintain an antigen-to-body-weight ratio consistent with that used in pigs, while ensuring adequate immune stimulation in the murine model. We have clarified this methodology in the revised Methods section. Please refer to the revised manuscript in lines 578-581.

9.Did authors test whether serum samples from commercial vaccine vaccinated animals reacted with the recombinant antigens (Fig. 3A)? What were the titers?

Response: Thank you for your insightful comment. We apologize for not specifying the components of the commercial vaccine earlier. The commercial vaccine used in this study is Porcilis® APP, which contains antigen components, including outer membrane proteins (OMPs) and Apx toxins (Apx I, Apx II, Apx III). However, it does not include GalT or GalU. Since our study focused on evaluating the immune response specifically against the recombinant rgalT-galU protein, which differs from the components present in the Porcilis® APP vaccine, we did not test whether serum samples from Porcilis® APP-vaccinated animals reacted with the recombinant antigens (rgalT-galU). However, in Figure 3B, we tested whether serum samples from Porcilis® APP-vaccinated animals reacted with three serotypes of APP antigens. We have now revised the manuscript to clarify this point and provide a more complete description of the Porcilis® APP vaccine components. Please refer to the revised manuscript, lines 588-589, for this update.

10.In the introduction the limitations of commercial vaccines were mentioned in the aspect of cross-protection. However, the data in Fig 3B shows that the commercial vaccine could induce a strong immune response against all the serotypes used in this study.

Response: Thank you for your thoughtful comment. We apologize for not clearly distinguishing between different types of vaccines in the introduction. The commercial vaccine we referenced in the introduction was meant to represent traditional inactivated vaccines, and we did not provide sufficient context regarding the background and efficacy data. Additionally, we failed to mention that Porcilis® APP is a subunit inactivated vaccine, which differs from traditional inactivated vaccines. Porcilis® APP contains antigen components, including outer membrane proteins (OMPs) and Apx toxins (Apx I, Apx II, Apx III), and it was able to induce a strong immune response against all the serotypes used in this study, as shown in Figure 3B. However, we also learned from consultations with pig farms that the commercial vaccine can induce significant stress responses, which is why we chose this vaccine as a control in the hope of improving vaccine efficacy in our study. We have revised the introduction to provide a more accurate description of the limitations of traditional inactivated vaccines and to clarify the nature of Porcilis® APP. Please refer to the revised manuscript in lines 75-93 for these updates. For more details about Porcilis® APP, please see lines 588-589.

11.9-1 what is the commercial vaccine serotype?

Response: Thank you for your question. We apologize for the oversight in not providing specific information regarding the serotype(s) of the commercial vaccine in the original manuscript. As mentioned earlier, Porcilis® APP is a subunit inactivated vaccine, which differs from traditional inactivated vaccines. Porcilis® APP contains antigen components, including outer membrane proteins (OMPs) and Apx toxins (Apx I, Apx II, Apx III). It is effective in preventing diseases caused by APP strains that express ApxI, ApxII, and/or ApxIII. We have revised the manuscript to provide a clearer description of Porcilis® APP and its components. Please refer to the revised manuscript, lines 588-589, for these updates.

12.I recommend analyzing the antibody titers for one antigen/vaccine between different serotypes (APP1, APP5b, APP7). This analysis may help to identify that new vaccine can induce strong immune response which does not differ between the strain while the commercial vaccine tends to induce stronger immune response to one strain but for the strain 2 it is significantly lower….

Response: Thank you very much for your valuable suggestion. Based on your advice, we have carefully revised and adjusted the results analysis and discussion to improve the quality of this manuscript. Specifically, we have now provided a comparison of antibody titers across the three APP serotypes (APP1, APP5b, and APP7) to better highlight the uniformity of the immune response elicited by the rgalT-galU vaccine as compared to the commercial vaccine. Please see the revised manuscript in lines 260-267.

Round 2

Reviewer 1 Report

Comments and Suggestions for Authors

The authors have modified the manuscript. 

Reviewer 2 Report

Comments and Suggestions for Authors

All comments have been addressed